# Development of a Machine Learning Framework to Aid Climate Model Assessment and Improvement: Case Study of Surface Soil Moisture

Francisco Andree Ramírez Casas *, Laxmi Sushama and Bernardo Teufel

Department of Civil Engineering, Trottier Institute for Sustainability in Engineering and Design,
McGill University, Montreal, QC H3A OC3, Canada
* Correspondence: francisco.ramirezcasas@mail.mcgill.ca

**Abstract:** The development of a computationally efficient machine learning-based framework to understand the underlying causes for biases in climate model simulated fields is presented in this study. The framework consists of a two-step approach, with the first step involving the development of a Random Forest (RF) model, trained on observed data of the climate variable of interest and related predictors. The second step involves emulations of the climate variable of interest with the RF model developed in step one by replacing the observed predictors with those from the climate model one at a time. The assumption is that comparing these emulations with that of a reference emulation driven by all observed predictors can shed light on the contribution of respective predictor biases to the biases in the climate model simulation. The proposed framework is used to understand the biases in the Global Environmental Multiscale (GEM) model simulated surface soil moisture (SSM) for the April–September period, over a domain covering part of north-east Canada. The grid cell-based RF model, trained on daily SSM and related climate predictors (water availability, 2 m temperature, relative humidity, snowmelt, maximum snow water equivalent) from the fifth generation European Centre for Medium-Range Weather Forecasts reanalysis (ERA5), demonstrates great skill in emulating SSM, with root mean square error of 0.036. Comparison of the five RF emulations based on GEM predictors with that based on ERA5 predictors suggests that the biases in the mean April–September SSM can be attributed mainly to biases in three predictors: water availability, 2 m temperature and relative humidity. The regions where these predictors contribute to biases in SSM are mostly collocated with the regions where they are shown to be the among the top three influential predictors through the predictor importance analysis, i.e., 2 m temperature in the southern part of the domain, relative humidity in the northern part of the domain and water availability over rest of the domain. The framework, without having to undertake expensive simulations with the climate model, thus successfully identifies the main causes for SSM biases, albeit with slightly reduced skill for heavily perturbed simulations. Furthermore, identification of the causes for biases, by informing targeted climate model improvements, can lead to additional reductions in computational costs.

**Keywords:** machine learning-based climate model validation; Random Forest; surface soil moisture; bias assessment; Eastern Canada

## 1. Introduction

Building credible climate models often require several experiments to identify the reasons for biases present in the simulated fields, and these experiments come at a great computational cost. For the emerging super-resolution/sub-km scale climate models, this cost increases exponentially. There has been a recent surge of interest in applying machine learning (ML) approaches to help reduce the computational cost of producing, validating, and extracting value from climate data [1]. For example, Schneider et al. [2] suggest that data driven models can replace parameterizations of unresolved scales, while stressing the importance of driving these models by governing physical equations when they are

known. Additionally, Reichstein et al. [3] identify other potential applications of ML in climate science for: (1) analysis of model-observation mismatch, which can be achieved by identifying and visualizing patterns of model error using ML so that targeted modifications can be conducted; (2) constraining sub-models by driving them with outputs from ML models, whereby uncertainty in model parameter calibration is reduced; and (3) surrogate modeling or emulation in order to increase speed of sensitivity analysis, model parameter calibration and deriving confidence intervals for estimated values.

Use of different types of climate emulators can be found in the literature ranging from simple statistical relations to more complex deep-learning models. Emulators are generally trained on a large sample of outputs available from previous simulations [4], such that their outputs lie appropriately and consistently within the distribution of the true simulation outputs, which can then be used in a computationally costly sensitivity analysis or for developing larger ensembles as required for uncertainty quantification. For example, Bellprat et al. [5] used a second order polynomial emulator that serves as a computationally cheap surrogate model to objectively calibrate a regional climate model, the COSMO-CLM. They report that the use of the emulator led to higher efficiency in the calibration process and the ability to sample a wider calibration space. Furthermore, they found the approach to be effective, as verified with a perfect model approach. Similarly, Castruccio et al. [6] used emulators to generate an ensemble of temperature and precipitation fields under arbitrary forcing scenarios based on a small set of precomputed runs from the Community Climate System Model. Temperature is captured with an infinite distributed lag model, in which current temperature is dependent on a weighted sum of past $\log(CO_2)$, while precipitation follows a regression model based on the response to changes in $CO_2$ and mean temperature. They argue that emulators of climate models can capture the full temporal dynamics of transient climate and emulated fields are produced instantly at negligible computational cost.

In Verrelst et al. [7], the authors developed a series of radiative transfer model (RTM) emulators using three regression algorithms, namely kernel ridge regression, neural networks and Gaussian processes regression, to perform global sensitivity analysis. Similar to GCMs, RTMs are computationally expensive, where up to tens of thousands of simulations are required to perform sensitivity analysis. They found the neural network and Gaussian processes regression emulators to be effective in approximating RTM outputs, which were then employed for extensive global sensitivity analysis.

An approach to downscale precipitation projections from GCM to regional climate model (RCM) resolutions using a convolutional autoencoder-based emulator is shown in Babaousmail et al. [8]. Their emulator was trained using 55 years (1951–2005) of historical rainfall data from eight GCMs from the Coupled Model Inter Comparison Project (CMIP5) as predictors and a dynamically downscaled precipitation product using the Rosby Centre regional atmospheric model (RCA4) as predictand. The emulator was then tested using RCA4 projections of future climate (2006–2100); they found that their emulator performs better at reproducing temporal precipitation projections than the mean ensemble of RCM products. However, it produced poor results when reproducing the spatial distribution of projected precipitation, which may be because their selected ML method does not address the spatial influences of precipitation.

More advanced machine learning methods and applications have emerged in recent years. For instance, Wu et al. [9] proposed a deep learning framework to generate super-resolution temperature and dew point fields for urban regions from coarse resolution regional climate model outputs. The deep learning model they employed consists of a sub-pixel convolution layer to generate high-resolution information from low resolution data, with adversarial training applied to improve physical consistency. For example, Teufel et al. [10] developed a framework that couples deep learning and physical modelling to produce high-resolution precipitation fields from low-resolution data, using a recurrent approach and considering physical processes. The generated precipitation estimates are temporally consistent and are able to recreate intense short-duration precipitation events. Their study takes from the existing work on video super-resolution, in which information

from multiple low-resolution frames is combined to estimate a high-resolution, with the addition of the antecedent high-resolution estimated frame as an input, so that information from previously estimated frames can be propagated to subsequent frames aiding the model in recreating fine details and temporally consistent fields. The wind field is used to derive the displacement of precipitation from one frame to the next, this corresponds to the fact that precipitation is advected by wind. The estimates of this method improve over time, as the framework is able to leverage past information to produce better estimates.

Assessment of climate model outputs using ML approaches is a developing area of research. In this study, an ML-based framework to understand the underlying sources of biases in climate model simulated fields using Random Forest (RF) powered emulators is developed and tested for the case of a regional climate model simulated surface soil moisture (SSM) over a domain covering part of north-east Canada. Such studies are severely lacking, and this study will form the basis for additional investigations. The rest of the paper is organized as follows: Section 2 describes the proposed ML-based framework, and datasets considered, including the climate model simulated fields. Results pertaining to the development of RF models for SSM emulation and application of the ML framework are presented in Section 3, followed by conclusions in Section 4.

## 2. Methodology

### 2.1. Machine Learning Framework

The proposed machine learning-based framework to identify mechanisms/variables responsible for biases in climate model simulated fields over a given region/domain is explained first in general terms, followed by its application for the cases considered in this study. As shown in Figure 1a, the framework consists of a two-step process. The first step involves the development of Random Forest (RF) models to emulate the variable of interest, i.e., the climate model simulated variable whose sources of biases need to be understood. To this end, observed gridded values of the variable of interest and climate predictors representing mechanisms that influence the variable of interest, along with geophysical fields (if required), need to be considered for training and validation. In the absence of gridded observed data, reanalysis products can be reasonable alternatives.

Random Forest (RF) is a machine learning algorithm which combines decision trees (DTs) and bootstrap-aggregation. The method allows for greater generalization over individual DTs, which tend to overfit training data [11]. Each tree is grown from a randomly selected sample (in-bag sample) with replacement of the original training data; the subset of left-out data is called the out-of-bag sample. The in-bag sample data is partitioned by a sequence of splits such that data is grouped recursively based on similar samples. A tree stops growing when no possible split will give both nodes the predetermined minimum number of elements, i.e., when the tree reached minimum leaf size (MLS). RF is widely used for both classification and regression applications. For the regression algorithm, the split of training data is performed at each node by finding the predictor that will minimize the mean squared error of the prediction made from the resulting samples at the child nodes. This process is repeated for the number of trees in the forest. The average estimate from all trees determined the RF prediction.

A key characteristic of the RF method is that it allows ranking of variables based on their predictive value, which is done using internal out-of-bag estimates [11]. For this, after each tree is constructed, the out-of-bag subset is randomly permuted and run through the corresponding tree. The regression values are stored for each predictor and once this process is completed, these values are compared against the true estimate. The importance values are determined based on the mean decrease in model accuracy after the random permutation. The decrease in accuracy is significant for important predictors, while the opposite is true for less important predictors.

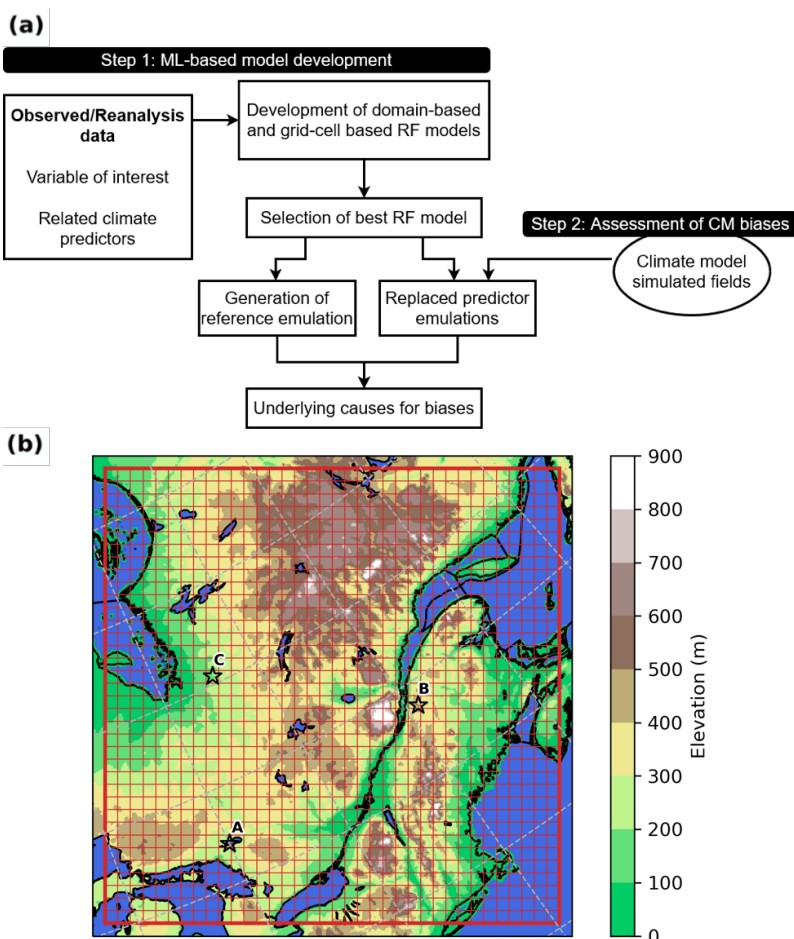

**Figure 1.** (**a**) Flow diagram of the proposed ML-based framework. (**b**) GEM experimental domain at 4 km; every 10th grid line is shown. A–C indicate locations selected for detailed analysis.

The RF model developed for this study can be domain-based (RF1) or grid cell-based (RF2), where, for the former, there is a single model trained on data from all grid cells covering the domain, while the latter is a collection of RF models for each grid cell. Following successful validation of the RF models, the best model can be selected, which is then used to generate the variable of interest based on observed predictors for the same duration as that of the climate model simulation, which will serve as the reference for the second step. In the next step, emulations of the variable of interest using the selected RF model are performed by replacing the observed predictors with those from the model, one at a time. Comparing these emulations of the variable of interest with that of the reference will help quantify the bias contribution of the respective predictor and its distribution over the domain. The key assumption is that the biases in the climate model predictors will propagate via the RF model and will be reflected in the corresponding emulation of the variable of interest, which can be quantified.

### 2.2. Case Study

In this study, the aim is to understand the biases in the regional climate model GEM (Global Environmental Multiscale) simulated surface soil moisture (SSM), at 4 km horizontal resolution, over a domain covering a part of eastern Canada (Figure 1b). GEM is used for numerical weather prediction by Environment and Climate Change Canada, it solves non-hydrostatic, deep atmosphere dynamics with an implicit, two-time-level semi-Lagrangian numerical scheme. Land processes in the model are represented using the Canadian Land-Surface Scheme (CLASS) version 3.5 [12]. The scheme includes prognostic equations for energy and water conservation and allows for a flexible number of ground layers and

thicknesses. In the GEM simulation setup, a 60 m deep configuration is used; it consists of 26 layers with the following distributions from the surface: 0.10, 0.20, 0.30, 0.40 m; the following 10 layers are 0.50 m each, 1.0, 3.0 m; and the last 10 layers are 5.0 m each.

For the purposes of this study, SSM is defined as liquid soil moisture content for the top 10 cm of the soil column. The focus is on the April–September period, i.e., when the top 10 cm is free of frozen water over most of the domain. For step 1, in the absence of observed gridded data, the fifth generation European Centre for Medium-Range Weather Forecasts (ECMWF) reanalysis ERA5, which provides hourly estimates of a large number of atmospheric, land and oceanic climate variables on a 30 km horizontal grid [13], is considered as the 'observed' data for SSM and its predictors.

The climate predictors considered in the development of both RF models are: relative humidity (RH), 2-meter temperature (TT), water availability (WA; defined here as precipitation minus evaporation), snowmelt (SMLT; defined as the difference of snow water equivalent from the previous time step, plus new snowfall) and maximum snow water equivalent (MSWE) from the preceding March. For the first four variables, 1- and 14-day lagged averages are considered. These predictors are selected as they influence SSM over the study region for the April–September period, with SMLT and MSWE being important for the April–May period, depending on the region.

Larger lags are not considered as the focus in this study in on-surface SM, which generally reflects the diurnal cycle and therefore only has short-term memory. In the case of RF1, in addition to climate predictors, topography (Figure 1b) and the contents of sand and clay in the top 10 cm of the soil column from the Harmonized World Soil Database v 1.2 are used in training of this model. The geophysical fields are not required in RF2 as separate models are developed for each grid cell. Thirty-two years of ERA5 data, spanning the 1989–2020 period, are used in this study, of which 26 years of data are used for training and the remaining 6 years are used for validation. The best model is selected based on RMSE and is used for generating the reference SSM (SSM_REF), driven by ERA5 predictors, for the same period as the GEM simulations.

Two GEM simulations are considered in Step 2 to test the framework. The first simulation, GEM1, uses normal geophysical fields, and is performed for the 1989–2020 period. The second simulation, GEM2, has perturbed geophysical fields, with the vegetation fraction replaced with bare soil; the simulation is run only for the years 1995–1996. The first year of results is not considered as it is used as spin up for the model to adjust to the absence of vegetation. The year 1996 was selected as the most representative of the entire data as it has the closest values of mean and standard deviation in comparison to the entire set. In Figure 2, the mean SSM values for ERA5 and both GEM simulations are shown. GEM2, with enhanced biases, is used to further test the robustness of the framework. GEM outputs for both simulations are re-gridded to the ERA5 grid using the nearest neighbor technique. The resulting analysis domain has 7,590 cells (110×69), of which 4176 have a water fraction above 60% and thus are not considered for either training or validation of the models.

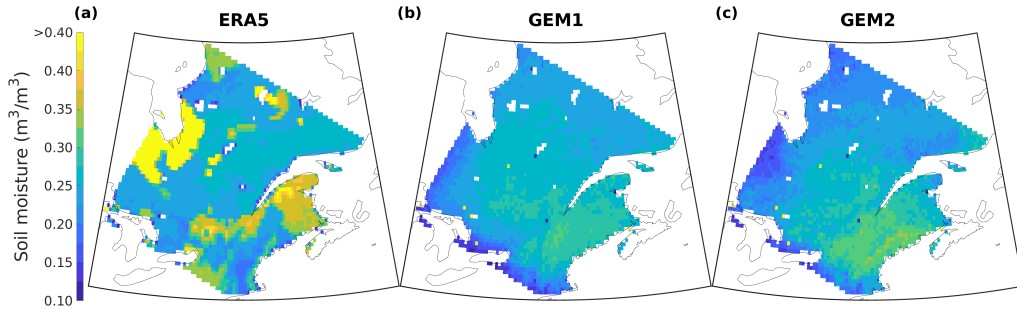

**Figure 2.** Mean surface soil moisture for the April–September period.(**a**) ERA5, (**b**) GEM1 and (**c**) GEM2. The former two are mean values for the 1989–2020 period while that for GEM2 is the mean for 1996.

Using the outputs of each GEM simulation, six emulations are performed with the selected RF model, by replacing WA, TT, RH, SMLT, MSWE and all predictors, respectively, of ERA5 with those from the GEM simulations; these emulations will be referred to as SSM_WA, SSM_TT, SSM_RH, SSM_SMLT, SSM_MSWE, SSM_ALL. Comparison of these with SSM_REF will enable quantification of the bias contribution of the respective predictor in the GEM simulations, i.e., the source and magnitude of biases in GEM.

## 3. Results

### 3.1. Development Of Random Forest Models

As discussed in Section 2, two RF models for SSM estimation are developed based on ERA5 data. RF1 is trained on data from all grid cells in the domain, while RF2, the grid cell-based model, is a collection of 3414 RF models. The RF1 model is thus trained with 33,791,316 daily values each of SSM and related predictors, with MLS of 1000, and 100 trees in the forest, which are found to be a good compromise between model performance and computational cost/effort as shown in Figure 3a; an MLS of 1000 yields RMSE of 0.042 $m^3\,m^{-3}$ and a reasonable model size of 17 GB. Each RF model in RF2 is trained with 4758 daily data of SSM and its nine predictors, and employs MLS of 1000 and 1000 trees for the forest (Figure 3b); MLS of 1000 yields average RMSE value of 0.036 $m^3\,m^{-3}$ and a total model size (i.e., sum of individual models) of 91 GB. These parameters result in $2\,m^{15}$ and $2\,m^2$ leaves per tree, respectively, for RF1 and RF2. Considering the range of SSM values that each model emulates, this is considered sufficient to capture the variability in the training dataset.

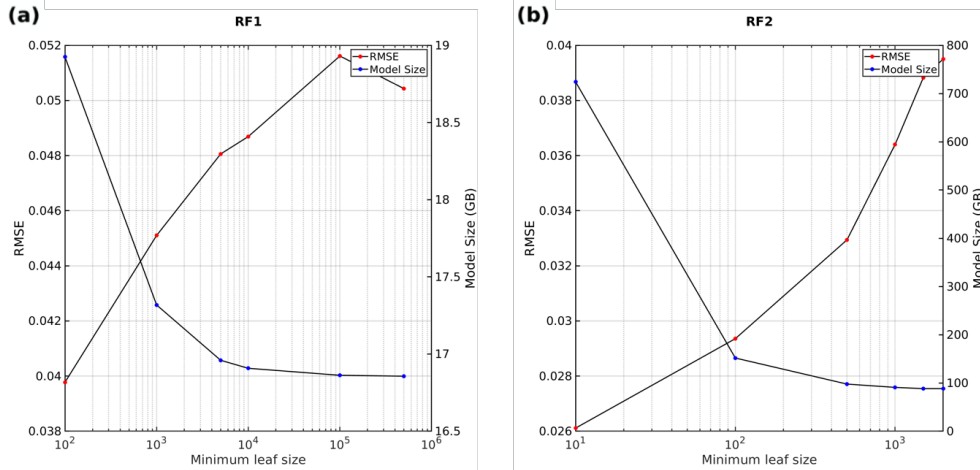

**Figure 3.** (**a**) Model performance for RF1, (**b**) RF2 (right) for different MLS values. Right-side axis indicates model file size; for RF2 this value corresponds to the sum of all models.

RF solves the overfitting problem of regression trees by aggregating predictions from a large number of trees and averaging them. Each tree is trained with slightly different randomly selected data (i.e., in-bag fraction feature) and therefore the prediction of individual trees varies. For the same reason, increasing the number of trees provides better results. For RF1, the value of leaves per tree is already high; a relatively small forest yields good emulation results, and no significant added value is found for forests containing over 100 trees. However, in the case of RF2, the leaves per tree is small, which would reduce the performance. This is addressed by having a larger number of trees, an order of magnitude higher than that for RF1. Again, no significant added value is found for forests with more than 1000 trees. Both RF1 and RF2 are trained using 90% of in-bag fraction. At each split, a third of all predictors are considered, and the selection of predictors is random at each node and the curvature test is selected as the feature selection algorithm.

RF1 and RF2 capture the spatial distribution of SSM as in the original ERA5 data; these results are shown in Figures 4 and 5, respectively, with RF2 showing slightly better

performance compared to RF1, which is expected as RF1 is a single model for the entire domain, while RF2 uses individual models for each grid cell. As can be seen from the SSM time series for three selected points A, B and C (with lowest, intermediate and highest mean SSM values, respectively) shown in Figure 6, the individual models that constitute RF2 are tasked with learning from a dataset with relatively low variability when compared with the dataset that RF1 is trained with. For instance, at point B, the model only learns from values in the range of 0.20 and 0.47 m$^3$ m$^{-3}$, with 50% of the data between the range of 0.35 and 0.41 m$^3$ m$^{-3}$. In the case of points A and C, the range of training data is even smaller, giving the models in RF2 the ability to predict to a high level of accuracy due to their site-specific learning. The dataset RF1 is trained on ranges from 0.04 to 0.77 m$^3$ m$^{-3}$. The large variability problem is solved by 'binning' SSM values with similar predictor features leading to reduced emulation capabilities, particularly for extreme values of SSM. The difference in performance between RF1 and RF2 is also reflected in the coefficients of determination, 0.88 and 0.99, respectively, for the training phase.

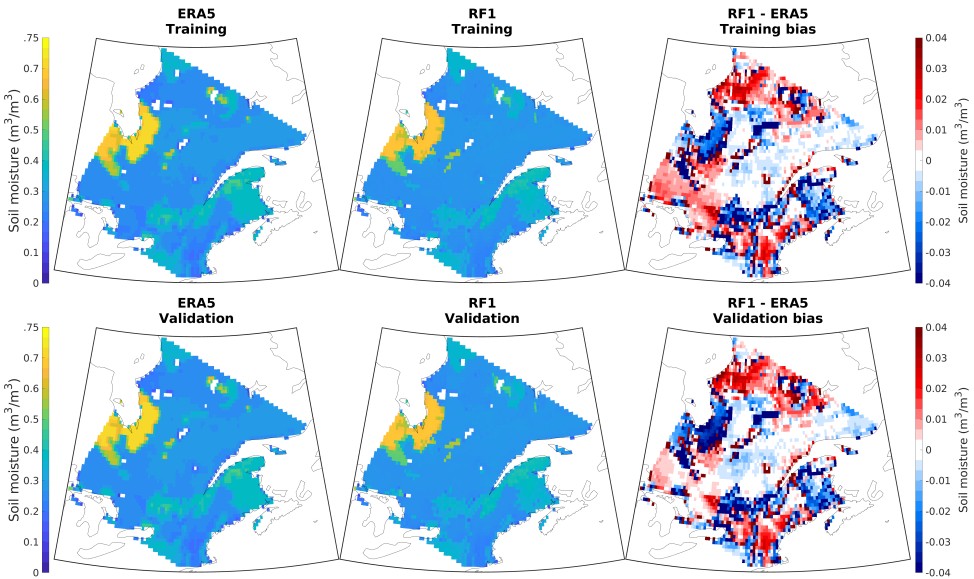

**Figure 4.** ERA5 (**left** column) and RF1 simulated (**center** column) SSM for training (**top** panel) and validation (**bottom** panel) periods and their biases (**right** column).

In terms of emulation bias, RF1 has considerable positive and negative bias across the domain (right column in Figure 4), a considerable portion of these biases are located in the extreme regions of the domain, where the model is unable to emulate SSM accurately. On the other hand, RF2 has a better ability to capture extreme values (Figure 5); mean validation bias follows the pattern of the difference between ERA5 training and validation data (figure not shown); if this difference is removed, a small negative bias is found in the emulated SSM of RF2; in the case of training bias, the values are negligible and appear to have a random nature; each model is able to capture accurately the training values, and this turns into a very small negative or positive bias.

The results from the relative importance analysis of the predictors for RF1 and RF2 are presented in Figure 7a. Unlike the importance analysis for RF1, which shows domain-wide importance, RF2 analysis studies predictor importance at grid cell level, which allows to better understand the spatial variability in predictor importance (shown in Figure 7b). Nevertheless, both models suggest WA as the most important variable, followed by topography for RF1 and RH for RF2. For RF1, sand and clay content fare similar in comparison to other climate variables. This is similar to the findings of Karthikeyan and Mishra [14], who reported soil characteristics to be very important at a depth of 5 cm using extreme gradient boosting over the United States, which is also consistent with Carranza

et al. [15], who found soil properties to have larger impact on their RF model emulation of root zone soil moisture, compared to many other climatic variables. As can be seen from Figure 7b, WA appears in the top three variables of importance across most of the domain. Temperature and relative humidity are also among the top three variables of importance, with temperature being mostly important for the southern regions, while relative humidity is important for the mid to northern regions. It is important to note that MWE and SMLT show a lower level of importance due to the fact that the RF models are trained to emulate the April–September SSM. RF models trained to simulate only April SSM may show different results with high importance for MSWE and SMLT.

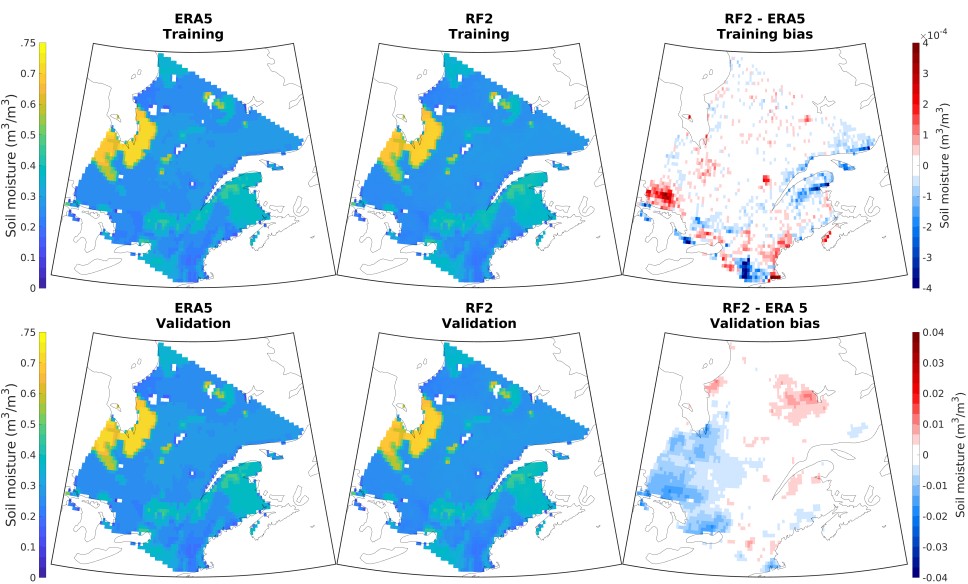

**Figure 5.** ERA5 (**left** column) and RF2 simulated (**center** column) SSM for training (**top** panel) and validation (**bottom** panel) periods and their biases (**right** column).

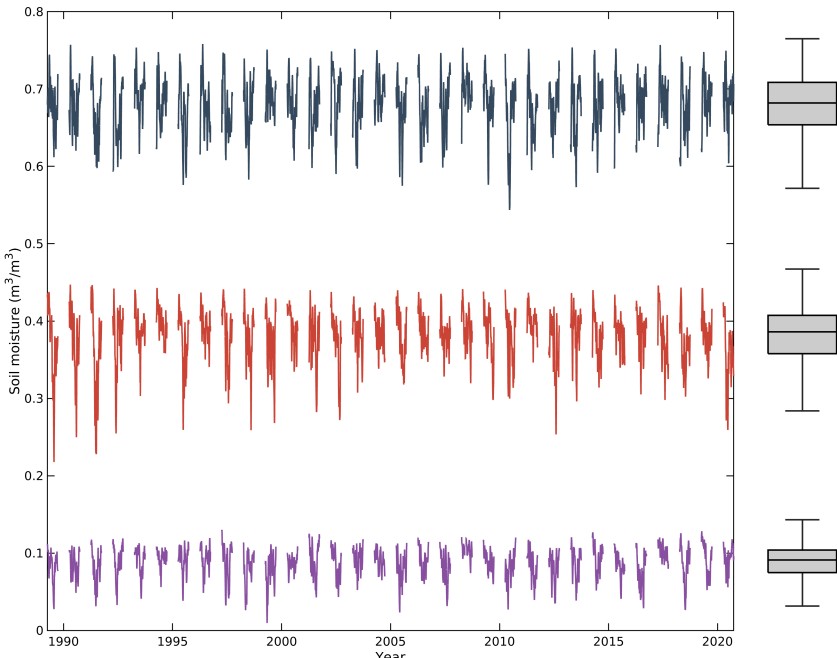

**Figure 6.** Evolution of SSM on the analysis points presented in Figure 1 (bottom: point A; middle: point B; top: point C). Seven-day moving averages are shown for the April–September period.

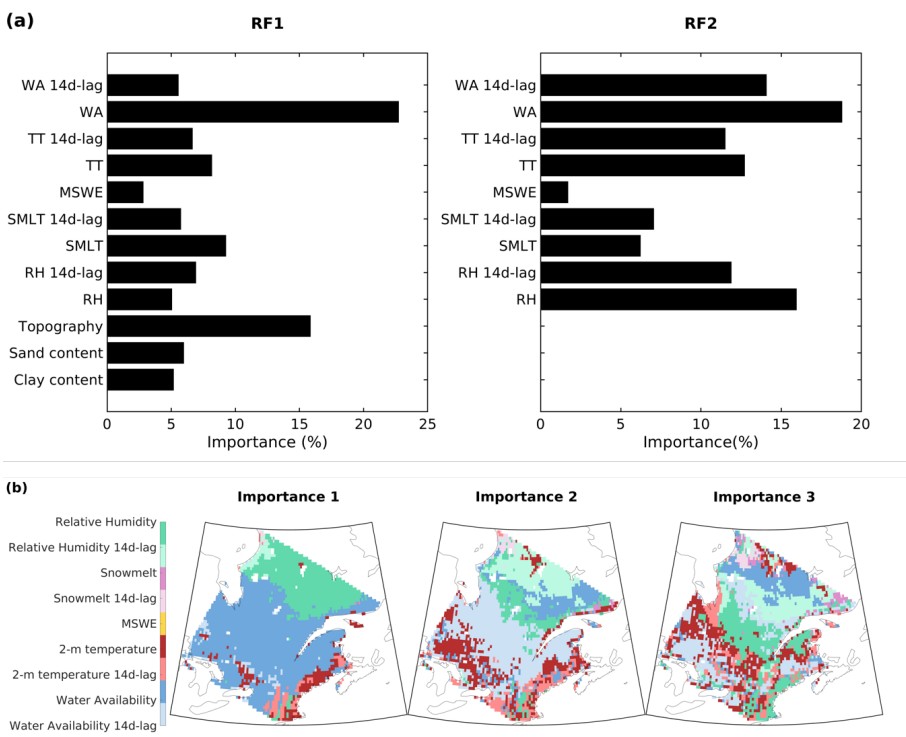

**Figure 7.** Mean surface soil moisture for the April–September period. (**a**) Relative importance of predictors for the RF1 model and average importance for the RF2 models. (**b**) Spatial view of the top 3 important variables for RF2. (WA: water availability; TT: 2 m temperature; MSWE: maximum snow water equivalent; SMLT: snowmelt; RH: relative humidity).

The last phase of the first step is to validate the RF models, for which six years of data are used. Results (bottom rows of Figures 4 and 5) suggest good performance with coefficients of determination being 0.81 and 0.96, respectively, for RF1 and RF2. Given the better performance of RF2, this model is selected for the next step of the framework. The reference SSM for the next step, SSM_REF, is generated using RF2 for the April to September period for the 1989–2020 period, which is the GEM1 simulation period. As discussed in Section 2, the second step of the framework consists of SSM emulations using RF2, but replacing the SSM predictors from the ERA5 dataset with those from GEM (SSM_WA, SS_RH, SSM_TT, SSM_SMLT, SSM_MSWE, SSM_ALL) with the goal of understanding the source of SSM biases. This is undertaken for both GEM1 (normal) and GEM2 (perturbed) simulations. SSM_ALL is the emulation with all ERA5 predictors replaced with those from GEM. The comparison of emulated SSM for the replaced GEM predictors with that of SSM_REF aids in finding the contribution of biases in the predictors to SSM biases in GEM.

### 3.2. GEM Simulations

### 3.2.1. Case 1: 'Normal' GEM Simulation

The GEM1 simulation is performed with geophysical fields set up to reproduce normal climate conditions. Six emulations are performed with RF2. Five of the six emulations take all but one predictor from ERA5 (SSM_WA, SSM_RH, SSM_TT, SSM_MSWE, SSM_SMLT). The sixth emulation (SSM_GEM) replaces all ERA5 predictors simultaneously with those from GEM1.

An initial finding is that when all inputs are replaced by GEM1 predictors, the RF2 model is able to reproduce surface soil moisture (with a slight overestimation) including spatial patterns across the entire domain, such as the high SSM of the wetlands region situated to the south of the Hudson Bay (see top panels in Figure 8). This is an interesting result when comparing it to the original GEM1 SSM field which does not present this

prominent regional feature (see Figure 2b). In GEM1 simulation, the SSM for the wetlands is low, as water infiltrates and/or evaporates due to the formulation used.

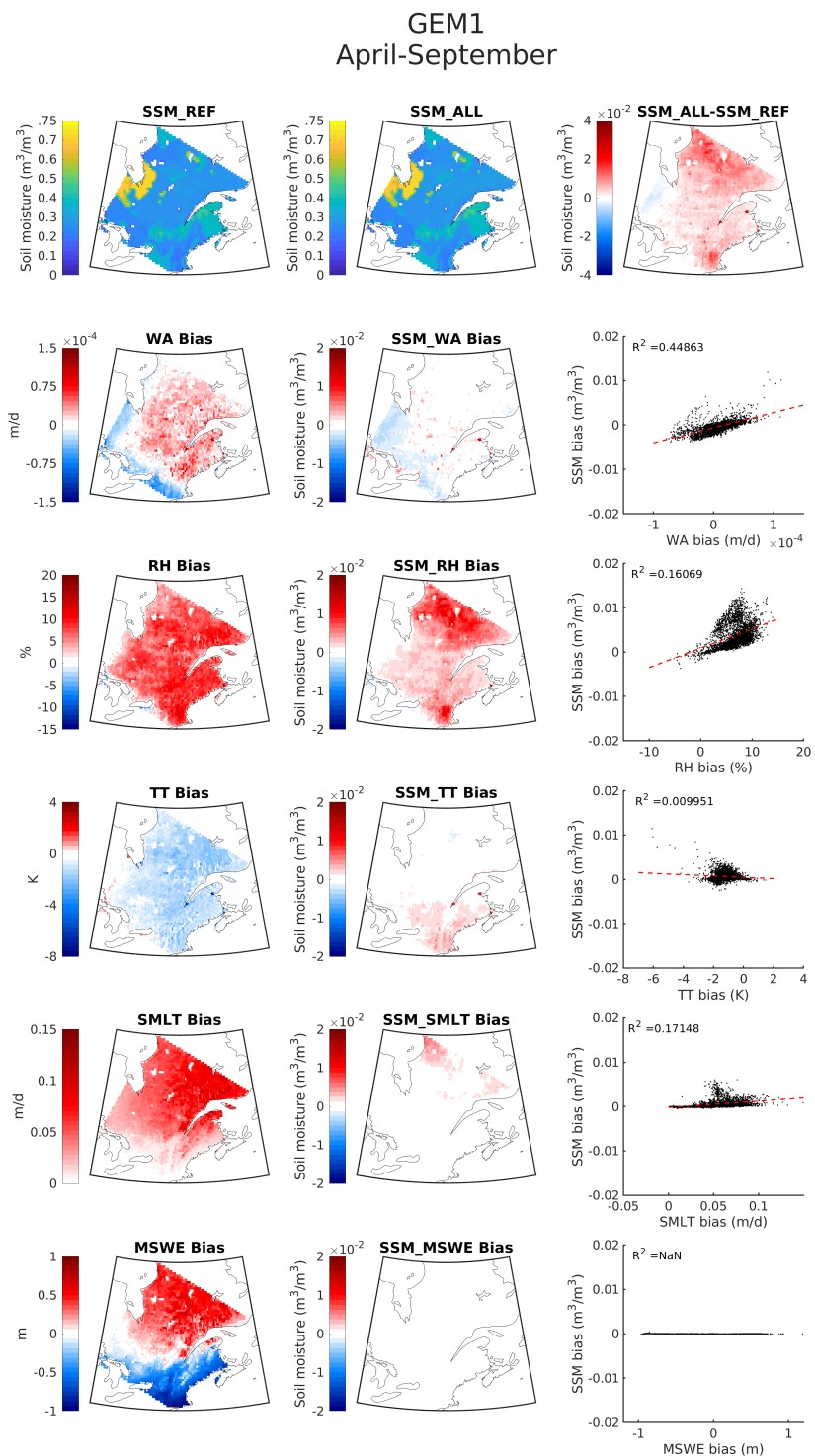

**Figure 8.** RF2 emulated SSM when using ERA5 predictors (**top left** panel), GEM1 predictors (**top center** panel), and their difference (**top right** panel) for the 1989−2020 period. GEM1 predictor biases when compared with ERA5 (**left** column). SSM biases when replacing ERA5 climate predictors with those from GEM1 in RF2. Scatter plots of SSM bias against predictor bias is presented in the right column.

The spatial patterns of the predictor biases are reflected in the respective SSM emulation, with the same or opposite sign. For instance, WA from GEM1 has a positive bias except for the south and western regions of the domain where there are negative biases. The spatial pattern of SSM differences between SSM_REF and SSM_WA are similar to those of WA biases, with similar signs (see row 2 of Figure 8). The same can be noted for RH (row 3 of Figure 8), which has a strong positive bias, which is reflected in the SSM biases (i.e., SSM_RH-SSM_REF). The positive correlation between the predictor (WA and RH) biases and biases in SSM emulations shown in the rightmost column of Figure 8 confirms this. In the case of TT (see row 4 of Figure 8), a negative bias in the predictor produces a positive bias in SSM emulation, reflected in the negative correlation between the biases, albeit low. As previously explained, the relative importance of MSWE and SMLT is small; therefore, their impact on the overall bias of the prediction is negligible. It must be noted that considering the biases for the shoulder months alone could yield different response as explained later in this section.

The positive correlation of biases for WA and RH are physically consistent as higher (lower) WA and RH translates into higher (lower) SSM. The same is true for TT as in general colder temperatures are conducive to reduced evaporation and therefore SMM via this mechanism. Overall, it can be noticed that the contribution of WA, RH and TT biases to SSM differences manifest in their respective regions of relative importance, as shown in Figure 7b. Besides, the sum of the SSM differences in different emulations more or less represent the difference between SSM_GEM1 and SSM_REF, suggesting that the predictor biases considered here can capture the reasons for the SSM biases in GEM well.

In Figure 9, the analysis for the month of April is presented; the same RF2 model developed for the April–September period is used. Although biases are similar in pattern to that for the April–September period, RH and SMLT biases are higher. SSM differences between the emulations and SSM_REF are clearly co-located with regions where the predictors have an important influence (as shown by the importance analysis in Figure 7b), even when considering a single month. Another noteworthy aspect is that, given the likely influence of SMLT on SSM during April, SMLT biases contribute to SSM differences between SSM_SMLT and SSM_REF for the northern regions of the domain.

### 3.2.2. Case 2: 'Perturbed' GEM Simulation

As mentioned in Section 2, the GEM2 simulation is designed to be differently biased than the 'normal' GEM simulation. The goal here is to test if the RF-based framework is able to identify the potential sources of errors in GEM2 simulated SSM. In Figure 10, it can be seen that the spatial patterns of the predictor biases are somewhat similar to that for the 'normal' simulation, except for WA and MSWE, and the biases in RH, TT and SMLT are also amplified. The difference between SSM_GEM2 and SSM_REF is also accordingly larger.

Results show that, even though the predictor values from GEM2 may lie outside of the range of the values used to train RF2, as is reflected by the large biases in RH and TT, the overall SSM differences (SSM_GEM2-SSM_REF) still appear to be the sum of the differences in individual SSM emulations with replaced predictors, with the largest contribution being that from RH and TT. This suggests that the RF-based framework demonstrates some skill even in this case in identifying underlying sources of errors. Examination of the scatter plots (right column of Figure 10), particularly for RH and TT, show considerable spread, and weaker association between SSM biases and those for WA, RH and TT, which were amongst the top three important predictors. This suggests that RF2 developed for more 'normal' conditions, as expected, may not yield definitive conclusions such as in the case of the highly perturbed GEM2 simulation considered here.

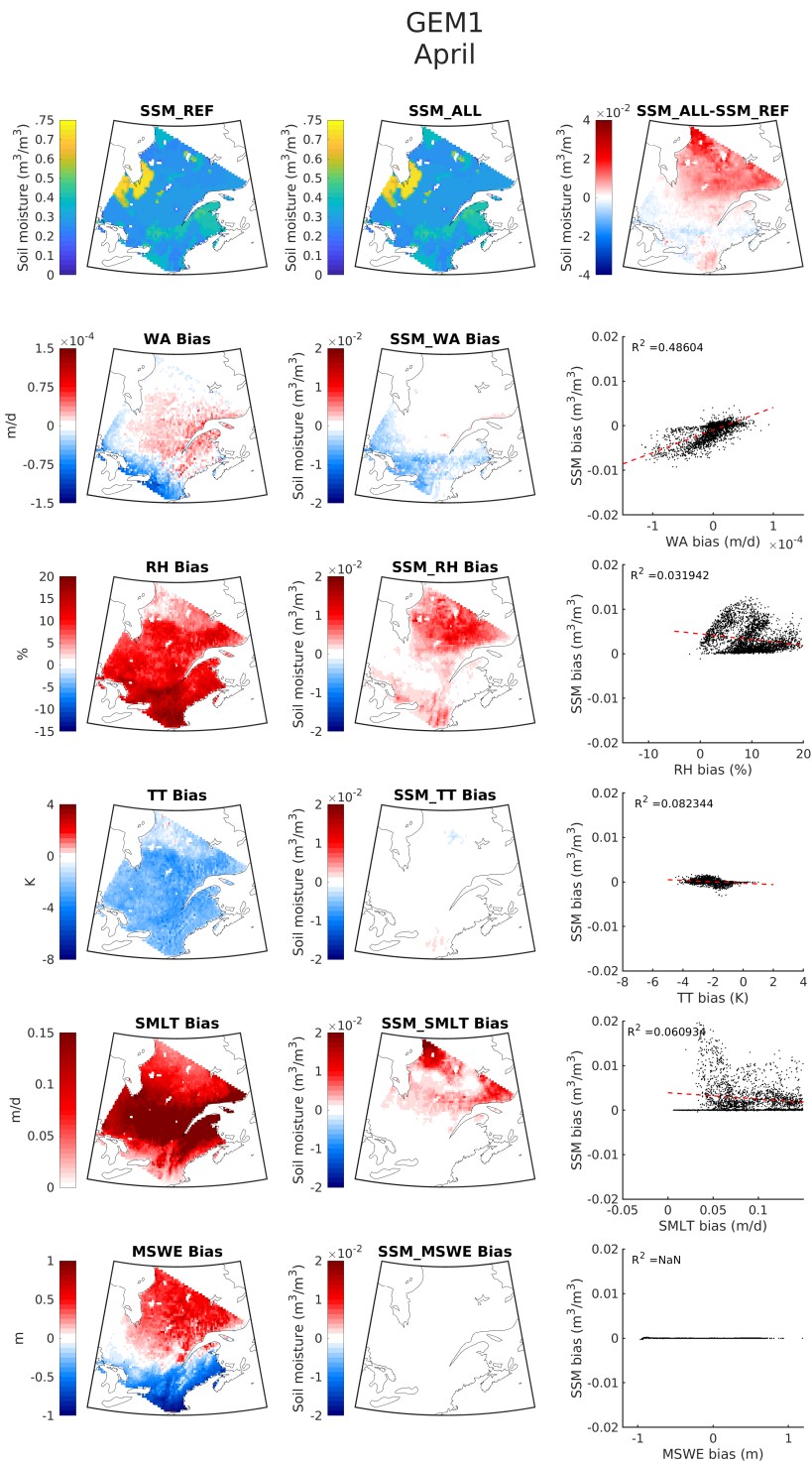

**Figure 9.** RF2 emulated SSM when using ERA5 predictors (**top left** panel), GEM1 predictors (**top center** panel) and their difference (**top right** panel) for the month of April in the 1989−2020 period. GEM1 predictor biases when compared with ERA5 (**left** column). SSM biases when replacing ERA5 climate predictors with those from GEM1 in RF2. Scatter plots of SSM bias against predictor bias are presented in the right column.

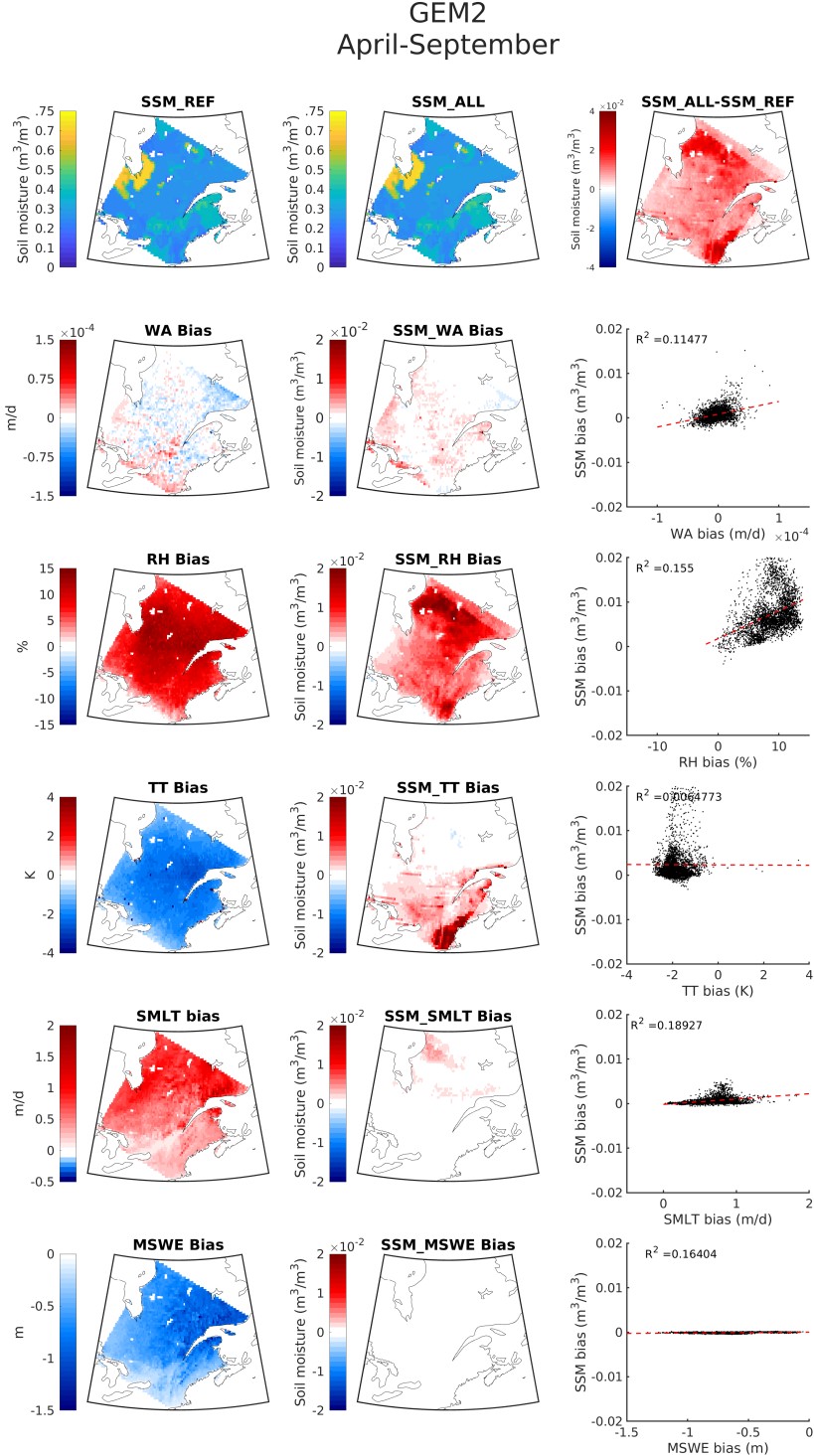

**Figure 10.** RF2 emulated SSM when using ERA5 predictors (**top left** panel), GEM2 predictors (**top center** panel) and their difference (**top right** panel) for the 1989−2020 period. GEM2 predictor biases when compared with ERA5 (**left** column). SSM biases when replacing ERA5 climate predictors with those from GEM2 in RF2. Scatter plots of SSM bias against predictor bias are presented in the right column.

## 4. Discussion and Conclusions

A new framework for identification of mechanisms responsible for biases in climate model simulated fields based on Random Forest models is proposed. The case study of GEM simulated surface soil moisture is presented.

Two sets of Random Forest models are developed and tested, one being a domain-based single model (RF1), and the other being an ensemble of separate models for each grid cell (RF2). Both models simulate SSM with high degree of accuracy, RF1 scores coefficients of determination of 0.81 and 0.88 for the validation and training datasets, respectively, while RF2 reaches 0.96 and 0.99 for the mentioned sets. RF2 is chosen over RF1 as it better captures the spatial and temporal variabilities of SSM. A reference emulation (SSM_REF) is computed using RF2 and all the predictors from the observation dataset (ERA5). Following, all predictors from the observed dataset are individually replaced with predictors obtained from GEM to emulate SSM. The contribution of bias is quantified by comparing the emulation of the individual replaced variables against SSM_REF.

Results show that water availability biases are mostly responsible for SSM biases, followed by relative humidity, and 2 m temperature biases. The development of a grid cell-based model allows to identify the regions where each predictor is most influential. This analysis shows that the regions where predictors contribute to biases in SSM are co-located with the regions where each is most influential. For the southern section of the domain, 2 m temperature biases are important, and for the northern portion relative humidity biases cause SSM bias. A second GEM simulation is performed with 'perturbed' geophysical fields in order to test the framework using heavily biased inputs, i.e., predictors outside of the predictor space used in training. When using these perturbed simulated fields, a reduction in the RF2 model skill to emulate SSM is observed.

The developed framework shows great skill in identifying the cause for biases; if coupled with targeted model improvements, it would lead to a significant reduction in computational costs.

One of the main limitations of this study is the mismatch of spatial resolutions between the observed ERA5 dataset at 30 km and the GEM simulated fields at 4 km. Future work could benefit from using ERA5-Land (9 km), as it is best to have the resolutions of observed data as close as the resolution from the climate model. Applying this framework to different variables and/or regions would require careful consideration concerning the selection of predictor variables. For instance, an investigation of soil moisture at deeper layers would require longer lags on the climate predictors, as the memory effect is more prevalent. In a similar manner, to use the framework in a different region an analysis of the main mechanisms would have to be conducted, e.g., the use of snowmelt in tropical regions would not be logical. Considering this, a method to systematize the predictor selection process could enhance the framework.

Although the RF models have good performance, parameter tuning was not exhaustive. Further gains in performance might be achieved following this line of research. Additionally, in this study two, arrangements of the training data were used in training the model, a single domain-based model and thousands of grid cell-based models. An alternative worth exploring is identifying regions for which soil moisture behaves similarly and developing a limited number of models based on these regions. Finally, a greater understanding of the framework's ability to handle highly perturbed data could be obtained by performing a longer GEM2 type simulation.

**Author Contributions:** Conceptualization, F.A.R.C., L.S. and B.T.; Formal analysis, F.A.R.C. and L.S.; Funding acquisition, L.S.; Investigation, F.A.R.C.; Methodology, F.A.R.C.; Software, F.A.R.C.; Supervision, L.S.; Validation, F.A.R.C.; Visualization, F.A.R.C.; Writing—original draft, F.A.R.C.; Writing—review & editing, L.S. and B.T. All authors have read and agreed to the published version of the manuscript.

**Funding:** This research was funded by the Canadian Space Agency grant number 21SUESDFIM.

**Data Availability Statement:** The data presented in this study are available on request from the corresponding author.

**Conflicts of Interest:** The authors declare no conflict of interest.

## Abbreviations

The following abbreviations are used in this manuscript:

| | |
|---|---|
| CLASS | Canadian Land Surface Scheme |
| COSMO-CLM | Cosmo-Climate Lokalmodell(German regional climate model) |
| CMIIP5 | Coupled Model Intercomparison Project Phase 5 |
| $CO_2$ | Carbon dioxide |
| DT | Decision Tree |
| ECMWF | European Centre for Medium-Range Weather Forecasts |
| ERA5 | ECMWF Re-Analysis |
| GCM | Global Climate Model |
| GEM | Global Environmental Multiscale |
| ML | Machine Learning |
| MLS | Minimum Leaf Size |
| MSWE | Maximum Snow Water Equivalent |
| RCA4 | Rosby Centre Regional Atmospheric Model Version 4 |
| RCM | Regional Climate Model |
| RF | Random Forest |
| RH | Relative Humidity |
| RMSE | Root Mean Square Error |
| RTM | Radiative Transfer Model |
| SM | Soil Moisture |
| SMLT | Snowmelt |
| SSM | Surface Soil Moisture |
| TT | 2 m Temperature |
| WA | Water Availability |

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
