# Peer review of "Development of a Machine Learning Framework to Aid Climate Model Assessment and Improvement: Case Study of Surface Soil Moisture"

_hydrology, doi:10.3390/hydrology9100186_

Round 1
Reviewer 1 Report
Please see attached comments

Reviewer 2 Report
This paper provides an interesting alternative to identify the error sources of GEMS. It is an exploration of the machine learning in the assessment of climate models. The scope of the paper is certainly within the journal. The paper is generally well written, however, the authors might consider certain improvement before it can be accepted for publication as it is.
1) The authors reviewed quite a bit on the climatic emulation with deep-learning models as it should be, but it would be more reasonable if the authors can mention about the development of bias identification in climate models.
2) It would be desirable if the authors can provide an illustration on the random forest (RF) algorithm. I'm not sure if it is a well-known method to the reader in hydrology community.
3) It is not clear to me why SMLT and MSWE were chosen as the predictors when basically they were only used in April (not the frozen soil situation was never discussed). For surface soil moisture, there are tons of other more relatable predictors, e.g., land cover/vegetation index, radiation, wind speed, soil texture etc. etc. I noticed the soil texture was indeed considered in RF1, but why it was not considered in RF2?
4) Line 300-302. Explain why the RF2 produces more spatial patterns than the original GEM1 SSM?
5) Line 304-306. Why the predictor bias and SSM emulations bias have same or opposite signs? From the scatter plots in Fig 8 and Fig 9, at least for WH and RH, they seem to be positively related?
6) Not sure if it is necessary to add the analysis of GCM2 as it really doesn’t add too much significant contribution. Instead, the authors might want consider some significant analysis on between the predictor errors and SSM emulation errors.
7) As mentioned before, the inclusion of SMLT and MSWE makes Fig 8, Fig 9 and Fig. 10 look redundant as they are apparently not affecting the summer SSM emulations.
Reviewer 3 Report
Development of a machine learning framework to aid climate model assessment and improvement: Case study of surface soil moisture:
· Add some of the most important quantitative results to the Abstract.
· Add/Replace the name of the study area with the Keywords.
· In the last paragraph of the Introduction, the authors should mention the weak point of former works (identification of the gaps) and describe the novelties of the current investigation to justify that the paper deserves to be published in this journal.
· Line 33, cite this useful paper on dealing with climate data to improve the literature and to show the importance of your work:
Calibration of mass transfer-based models to predict reference crop evapotranspiration
· Lines 101-105 are not necessary and should be deleted.
· The figure captions should be independent.
· Discus the main reasons for the variations of the mean surface soil moisture for the April-September period.
· Why did you peak ERA5 among climate reanalysis models (e.g., ERA5-Land, MERRA-2, JRA-55, etc.)?
· How can expand the results to other regions with similar/different climates?
Round 2
Reviewer 1 Report
It is highly recommended that renalysis data must be validated against in-situ measurements before they can be used as "observed data" in this study
Reviewer 3 Report
Acceptable.